# Reinforcement Learning in Factored MDPs: Oracle-Efficient Algorithms and Tighter Regret Bounds for the Non-Episodic Setting

**Ziping Xu**
Department of Statistics
University of Michigan
zipingxu@umich.edu

**Ambuj Tewari**
Department of Statistics
University of Michigan
tewaria@umich.edu

## Abstract

We study reinforcement learning in non-episodic factored Markov decision processes (FMDPs). We propose two near-optimal and oracle-efficient algorithms for FMDPs. Assuming oracle access to an FMDP planner, they enjoy a Bayesian and a frequentist regret bound respectively, both of which reduce to the near-optimal bound $\widetilde{O}(DS\sqrt{AT})$ for standard non-factored MDPs. We propose a tighter connectivity measure, factored span, for FMDPs and prove a lower bound that depends on the factored span rather than the diameter $D$. In order to decrease the gap between lower and upper bounds, we propose an adaptation of the REGAL.C algorithm whose regret bound depends on the factored span. Our oracle-efficient algorithms outperform previously proposed near-optimal algorithms on computer network administration simulations.

## 1 Introduction

Designing computationally and statistically efficient algorithms is a core problem in Reinforcement Learning (RL). There is a rich line of works that achieve a strong sample efficiency guarantee with regret analysis in tabular MDPs, where state and action spaces are finite and small (Jaksch et al., 2010; Osband et al., 2013; Dann and Brunskill, 2015; Kearns and Singh, 2002). A current challenge in RL is dealing with large state and action spaces where even polynomial dependence of regret on state and action spaces size is unacceptable. One idea to meet this challenge is to consider MDPs with compact representations. For example, factored MDPs (FMDPs) (Boutilier et al., 2000) represent transition functions of MDPs using a compact Dynamic Bayesian network (DBN) (Ghahramani, 1997). FMDPs have a variety of applications in important real-world problems, e.g. multi-agent RL, and they also serve as important case studies in theoretical RL research (Guestrin et al., 2002a,c; Tavakol and Brefeld, 2014; Sun et al., 2019).

There is no FMDP planner that is both computationally efficient and accurate (Goldsmith et al., 1997; Littman, 1997). Guestrin et al. (2003) proposed approximate algorithms with prespecified basis functions and bounded approximation errors. For the even harder online learning setting, we study *oracle-efficient algorithms*, which learn an unknown FMDP efficiently when given access to a planning oracle. In this paper, our goal is to design efficient online algorithms that only make a polynomial number of calls to the oracle planning oracle. Side-stepping the computational intractability of the offline problem by assuming oracle access to a solver has yielded insights into simpler decision making problems. For example, oracle-based efficient algorithms have been proposed for the contextual bandit problem (Syrgkanis et al., 2016; Luo et al., 2018).

Online learning in *episodic* FMDP has been studied by Osband and Van Roy (2014). They proposed two algorithms, PSRL (Posterior Sampling RL) and UCRL-factored with near-optimal Bayesian and

| Algorithms | Works | Regret bounds | Oracle |
|---|---|---|---|
| F-RMAX | Strehl et al. (2007) | (mixing rate) $T^{3/4}$ | Planning on FMDP |
| UCRL-Factored | Osband and Van Roy (2014) | $DT^{1/2}$ | Planning on Bounded FMDP |
| PSRL | This work | $DT^{1/2}$ (Bayesian) | Planning on FMDP |
| DORL | This work | $DT^{1/2}$ | Planning on FMDP |
| FSRL | This work | $QT^{1/2}$ | Optimizing average reward with bounded factored span |

Table 1: Improvements of our methods over previous results on non-episodic FMDP in terms of regret bounds. There are three types of oracle: 1) Planning on a known FMDP; 2) Planning on a known bounded FMDP; 3) Optimizing average rewards over a confidence set with bounded factored span. As far as we known, the first oracle has multiple efficient approximate solutions, while the last two do not. Of all the algorithms, FSRL gives the tightest bound.

frequentist regret bounds, respectively. Their UCRL-factored algorithm relies on solving a Bounded FMDP (Givan et al., 2000), which is an even stronger assumption than the access to a planning oracle. Recently, Tian et al. (2020) improved the bound using a Bernstein-style confidence set.

This work studies FMDPs in the more challenging *non-episodic* setting. Previous studies in non-episodic FMDPs either have some high order terms in their analysis (Strehl, 2007) or depend on some strong connectivity assumptions, e.g. mixing time (Kearns and Koller, 1999). There is no near-optimal regret analysis in this setting yet.

Regret analysis in the non-episodic setting relies on the connectivity assumptions. Previous available connectivity assumptions include *mixing time* (Lewis and Puterman, 2001), *diameter* (Jaksch et al., 2010) and *span of bias vector* (Bartlett and Tewari, 2009). Mixing time is the strongest assumption and span of bias vector gives the tightest regret bound among the three. However, we show that even upper bound using span can be loose if the factor structure is not taken into account.

This paper makes three main contributions:

1. We provide two oracle-efficient algorithms, DORL (Discrete Optimism RL) and PSRL (Posterior Sampling RL), with near-optimal frequentist regret bound and Bayesian regret bound respectively. Both upper bounds depend on the *diameter* of the unknown FMDP. The algorithms call the FMDP planner only a polynomial number of times. The upper bound of DORL, when specialized to the standard non-factored MDP setting, matches that of UCRL2 (Jaksch et al., 2010). The same applies to the upper bound of PSRL in the non-factored setting (Ouyang et al., 2017).
2. We propose a tighter connectivity measure especially designed for FMDPs, called *factored span* and prove a regret lower bound that depends on the *factored span* of the unknown FMDP rather than its *diameter*.
3. Our last algorithm FSRL is not oracle efficient but its regret scales with factored span, and using it, we are able to *reduce the gap between upper and lower bounds on regret* in terms of both the dependence on diameter and on $m$, the number of factors.

## 2 Preliminaries

We first introduce necessary definitions and notation for non-episodic MDPs and FMDPs.

### 2.1 Non-episodic MDP

We consider a setting where a learning agent interacts without resets or episodes with a Markov decision process (MDP), represented by $M = \{\mathcal{S}, \mathcal{A}, P, R\}$, with finite state space $\mathcal{S}$, finite action space $\mathcal{A}$, the transition probability $P \in \mathcal{P}_{\mathcal{S} \times \mathcal{A}, \mathcal{S}}$ and reward distribution $R : \mathcal{P}_{\mathcal{S} \times \mathcal{A}, [0,1]}$. Here $\Delta(\mathcal{X})$ denotes a distribution over the space $\mathcal{X}$. Let $\mathcal{G}(\mathcal{X})$ be the space of all possible distributions over $\mathcal{X}$ and $\mathcal{P}_{\mathcal{X}_1, \mathcal{X}_2}$ is the class of all the mappings from space $\mathcal{X}_1$ to $\mathcal{G}(\mathcal{X}_2)$. Let $S := |\mathcal{S}|$ and $A := |\mathcal{A}|$.

An MDP $M$ and a learning algorithm $\mathcal{L}$ operating on $M$ with an arbitrary initial state $s_1 \in \mathcal{S}$ constitute a stochastic process described by the state $s_t$ visited at time step $t$, the action $a_t$ chosen by

$\mathcal{L}$ at step $t$, the reward $r_t \sim R(s_t, a_t)$ and the next state $s_{t+1} \sim P(s_t, a_t)$ obtained for $t = 1, \ldots, T$. Let $H_t = \{s_1, a_1, r_1, \ldots, s_{t-1}, a_{t-1}, r_{t-1}\}$ be the trajectory up to time $t$.

Below we will define our regret measure in terms of undiscounted sum of rewards. To derive non-trivial upper bounds, we need some connectivity constraint. There are several subclasses of MDPs corresponding to different types of connectivity constraints (e.g., see the discussion in Bartlett and Tewari (2009)). We first focus on the class of *communicating* MDPs, i.e., the diameter of the MDP, which is defined below, is upper bounded by some $D < \infty$.

**Definition 1** (Diameter). *Consider the stochastic process defined by a stationary policy $\pi : \mathcal{S} \to \mathcal{A}$ operating on an MDP $M$ with initial state $s$. Let $T(s' \mid M, \pi, s)$ be the random variable for the first time step in which state $s'$ is reached in this process. Then the diameter of $M$ is defined as*

$$D(M) := \max_{s \neq s' \in \mathcal{S}} \min_{\pi : \mathcal{S} \to \mathcal{A}} \mathbb{E}\left[T\left(s' | M, \pi, s\right)\right].$$

A stationary policy $\pi$ on an MDP $M$ is a mapping $\mathcal{S} \mapsto \mathcal{A}$. An average reward (also called gain) of a policy $\pi$ on $M$ with an initial distribution $s_1$ is defined as

$$\lambda(M, \pi, s_1) = \limsup_{T \to \infty} \frac{1}{T} \mathbb{E}\left[\sum_{t=1}^{T} r(s_t, \pi(s_t))\right],$$

where the expectation is over trajectories $H_T$. We restrict the choice of policies within the set $\Pi$ of all policies whose average reward is independent of the starting state $s_1$. It can be shown that for a communicating MDP the optimal policies with the highest average reward are in the set and neither the optimal policy nor the optimal reward depends on the initial state. Let $\pi(M) = \arg\max_{\pi \in \Pi} \lambda(M, \pi, s_1)$ denote the optimal policy for MDP $M$ and $\lambda^*(M)$ denote the optimal average reward or optimal gain. For any optimal policy $\pi(M)$, following the definition in Puterman (2014), we define

$$h(M, s) = \mathbb{E}\left[\sum_{t=1}^{\infty} (r_t - \lambda^*(M)) \mid s_1 = s\right], \text{ for } s = 1, \ldots S,$$

where the expectation is taken over the trajectory generated by policy $\pi(M)$. And the bias vector of MDP $M$ is $\boldsymbol{h}(M) := (h(M, 1), \ldots, h(M, S))^T$. Let the span of a vector $\boldsymbol{h}$ be $sp(\boldsymbol{h}) := \max_{s_1, s_2} \boldsymbol{h}(s_1) - \boldsymbol{h}(s_2)$. Note that if there are multiple optimal policies, we consider the policy with the largest span for its bias vector.

We define the regret of a reinforcement learning algorithm $\mathcal{L}$ operating on MDP $M$ up to time $T$ as

$$R_T := \sum_{t=1}^{T} \left(\lambda^*(M) - r_t\right),$$

and Bayesian regret w.r.t. a prior distribution $\phi$ on a set of MDPs as $\mathbb{E}_{M \sim \phi} R_T$.

**Optimality equation for average reward criterion.** We let $\boldsymbol{R}(M, \pi)$ denote the $S$-dimensional vector with each element representing $\mathbb{E}_{r \sim R(s, \pi(s))}[r]$ and $P(M, \pi)$ denote the $S \times S$ matrix with each row as $P(s, \pi(s))$. For any communicating MDP $M$, the bias vector $\boldsymbol{h}(M)$ satisfies the following equation (Puterman, 2014):

$$\mathbf{1}\lambda^*(M) + \boldsymbol{h}(M) = \boldsymbol{R}(M, \pi^*) + P(M, \pi^*)\boldsymbol{h}(M). \tag{1}$$

## 2.2 Factored MDP

Factored MDP is modeled with a DBN (Dynamic Bayesian Network) (Dean and Kanazawa, 1989), where transition dynamics and rewards are factored and each factor only depends on a finite scope of state and action spaces. We use the definition in Osband and Van Roy (2014). We call $\mathcal{X} = \mathcal{S} \times \mathcal{A}$ factored set if it can be factored by $\mathcal{X} = \mathcal{X}_1 \times \ldots \times \mathcal{X}_n$. Note this formulation generalizes those in Strehl (2007); Kearns and Koller (1999) to allow the factorization of the action space as well. A visual illustration is shown in Figure 2.2.

**Definition 2** (Scope operation for factored sets). *For any subset of indices $Z \subseteq \{1, 2, .., n\}$, let us define the scope set $\mathcal{X}[Z] := \bigotimes_{i \in Z} \mathcal{X}_i$. Further, for any $x \in \mathcal{X}$ define the scope variable $x[Z] \in \mathcal{X}[Z]$ to be the value of the variables $x_i \in \mathcal{X}_i$ with indices $i \in Z$. For singleton sets $\{i\}$, we write $x[i]$ for $x[\{i\}]$ in the natural way.*

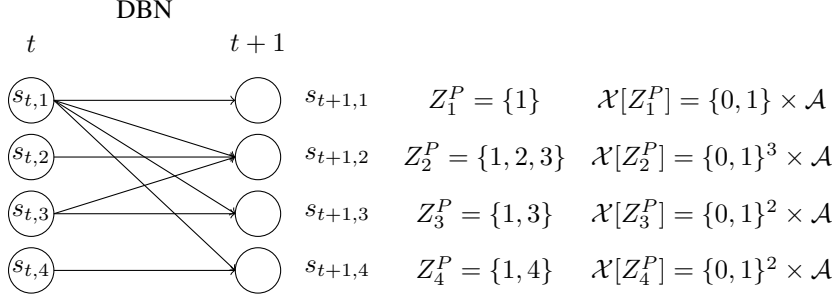

Figure 1: An example FMDP with 4 factors. Each node represents a dimension on a four dimensional vector. Each dimension is from $\{0, 1\}$. Arrows represents the dependence between two state vectors at step $t$ and $t + 1$. In this example, all four dimensions depend on dimension 1 and themselves.

**Definition 3** (Factored reward distribution). *A reward distribution $R$ is factored over $\mathcal{X}$ with scopes $Z_1^R, \ldots, Z_l^R$ if and only if, for all $x \in \mathcal{X}$, there exist distributions $\left\{ R_i \in \mathcal{P}_{\mathcal{X}[Z_i^P],[0,1]} \right\}_{i=1}^{l}$ such that any $r \sim R(x)$ can be decomposed as $\sum_{i=1}^{l} r_i$, with each $r_i \sim R_i(x[Z_i^R])$ individually observable. Throughout the paper, we also let $R(x)$ denote reward function of the distribution $R(x)$, which is the expectation $\mathbb{E}_{r \sim R(x)}[r]$.*

**Definition 4** (Factored transition probability). *A transition function $P$ is factored over $\mathcal{S} \times \mathcal{A} = \mathcal{X}_1 \times \ldots \times \mathcal{X}_n$ and $\mathcal{S} = \mathcal{S}_1 \times \ldots \mathcal{S}_m$ with scopes $Z_1^P, \ldots, Z_m^P$ if and only if, for all $x \in \mathcal{X}, s \in \mathcal{S}$ there exists some $\left\{ P_i \in \mathcal{P}_{\mathcal{X}[Z_i],\mathcal{S}_i} \right\}_{i=1}^{m}$ such that,*

$$P(s|x) = \prod_{i=1}^{m} P_i \left( s[i] \mid x\left[Z_i\right] \right).$$

*For simplicity, let $P(x)$ also denote the vector for the probability of each next state from current pair $x$. We define $P_i(x)$ in the same way.*

**Assumptions on FMDP.** To ensure a finite number of parameters, we assume that $|\mathcal{X}[Z_i^R]| \leq L$ for $i \in [n]$, $|\mathcal{X}[Z_i^P]| \leq L$ for $i \in [m]$ and $|\mathcal{S}_i| \leq W$ for all $i \in [m]$ for some finite $L$ and $W$. Furthermore, we assume that $r \sim R$ is in $[0, 1]$ with probability 1.

**Empirical estimates.** We first define number of visits for each factored set. Let $N_{R_i}^t(x) :=$ $\sum_{\tau=1}^{t-1} \mathbb{1}\{x_\tau[Z_i^R] = x\}$ be the number of visits to $x \in \mathcal{X}[Z_i^R]$ until $t$, $N_{P_i}^t(x)$ be the number of visits to $x \in \mathcal{X}[Z_i^P]$ until $t$ and $N_{P_i}^t(s, x)$ be the number of visits to $x \in \mathcal{X}[Z_i^P], s \in \mathcal{S}_i$ until $t$. The empirical estimate for $R_i(x)$ is $\hat{R}_i^t(x) = \sum_\tau^{t-1} r_\tau \mathbb{1}\{x_\tau[Z_i^R] = x\}/\max\{1, N_{R_i}^t(x)\}$ for $i \in [l]$. Estimate for transition probability is $\hat{P}_i^t(s \mid x) = \frac{N_{P_i}^t(s,x)}{\max\{1,N_{P_i}^t(x)\}}$ for $i \in [m]$. Algorithms for non-episodic MDP typically proceed in internal episodes, where no reset takes place at the end of an episode. We let $N_{R_i}^k, \hat{R}_i^k$ and $\hat{P}_i^k$ be $N_{R_i}^{t_k}, \hat{R}_i^{t_k}$ and $\hat{P}_i^{t_k}$ with $t_k$ being the first step of episode $k$.

## 3 Oracle-efficient Algorithms

We use PSRL (Posterior Sampling RL) and a modified version of UCRL-factored, called DORL (Discrete Optimism RL). The main idea of DORL is the simple fact that keeping a discrete set of MDPs reflecting the extreme points in the confidence set is sufficient for optimism. *By keeping a discrete set, one can avoid the hard computational oracle of planning on a bounded FMDP.*

Both PSRL and DORL use a fixed policy within an episode. For PSRL, we apply optimal policy for an MDP sampled from the posterior distribution of the true MDP. For DORL, instead of optimizing over a bounded MDP, we construct a new extended MDP, which is also factored with the number of parameters polynomial in that of the true MDP. Then the optimal policy of the extended FMDP is mapped to the policy space of the true FMDP. Instead of using dynamic episodes, we show that a

simple fixed episode scheme can also give us near-optimal regret bounds. The fixed episode scheme is proposed to ease the technical difficulty on the PSRL. Ouyang et al. (2017) combines doubling trick (Jaksch et al., 2010) and constraining that $T_{k+1} \le T_k + 1$, which gives the same regret bound. However, this scheme changes episodes slower, which could be more efficient in practice. Algorithm details are shown in Algorithm 1.

## 3.1 Extended FMDP

**Previous two constructions.** Previous near-optimal algorithms on regular MDP depend on constructing an extended MDP with a high probability of being optimistic. Jaksch et al. (2010) constructs the extended MDP with a continuous action space to allow choosing any transition probability in a confidence set. This construction generates a bounded-parameter MDP. Agrawal and Jia (2017) instead sample MDPs only from the extreme points of a similar confidence set and combine them by adding an extra dimension on action space to decide which MDP to use.

Solving the bounded-parameter MDP with the first construction, which requires storing and ordering the $S$-dimensional bias vector, is not feasible for FMDPs. There is no direct adaptation that mitigates this computation issue. We show that the second construction using only a discrete set of MDPs can be solved with a much lower complexity in the FMDP setting.

We formally describe the construction. For simplicity, we ignore the notations for $k$ in this section. First define the error bounds as an input. For every $x \in \mathcal{X}[Z_i^P]$, $s \in \mathcal{S}$, we have an error bound $E_{P_i}(s \mid x)$ for transition probability $\hat{P}_i(s \mid x)$. For every $x \in \mathcal{X}[Z_i^R]$, we have an error bound $E_{R_i}(x)$ for $\hat{R}_i(x)$. At the start of episode $k$ the construction takes the inputs of $\hat{M}_k$ and the error bounds, and outputs the extended MDP $M_k$.

**Extreme transition dynamic.** We first define the extreme transition probability mentioned above in factored setting. Let $P_i(x)^{s+}$ be the transition probability that encourages visiting $s \in \mathcal{S}_i$, denoted by

$$P_i(x)^{s+} = P_i(x) - E_{P_i}(\cdot \mid x) + \mathbb{1}_s \sum_j E_{P_i}(j \mid x),$$

where $\mathbb{1}_j$ is the vector with all zeros except for a one on the $j$-th element. By this definition, $P_i(x)^{s+}$ is a new transition probability that puts all the uncertainty onto the direction $s$. An example is shown in Figure 2. Our construction assigns an action for each extreme transition dynamic.

$$\underbrace{\begin{pmatrix} 0.5 \\ 0.3 \\ 0.2 \end{pmatrix}}_{\text{Estimated dynamic}} - \underbrace{\begin{pmatrix} 0.1 \\ 0.05 \\ 0.05 \end{pmatrix}}_{\text{Uncertainty}} + \underbrace{\begin{pmatrix} 0 \\ 0.2 \\ 0 \end{pmatrix}}_{\text{Encourage visiting } s_2} = \underbrace{\begin{pmatrix} 0.4 \\ 0.45 \\ 0.15 \end{pmatrix}}_{\text{Extreme dynamic}}$$

Figure 2: An extreme transition dynamic that encourages visiting the second state out of three states.

**Construction of extended FMDP.** Our new factored MDP is denoted by $M_k = \{\mathcal{S}, \tilde{\mathcal{A}}, \tilde{P}, \tilde{R}\}$, where $\tilde{\mathcal{A}} = \mathcal{A} \times \mathcal{S}$ and the new scopes $\{\tilde{Z}_i^R\}_{i=1}^l$ and $\{\tilde{Z}_i^P\}_{i=1}^m$ are the same as those for the original MDP.

Let $\tilde{\mathcal{X}} = \mathcal{X} \times \mathcal{S}$. The new transition probability is factored over $\tilde{\mathcal{X}} = \bigotimes_{i \in [m]} (\mathcal{X}[Z_i^P] \times \mathcal{S}_i)$ and $\mathcal{S} = \bigotimes_{i \in [m]} \mathcal{S}_i$ with the factored transition probability to be

$$\tilde{P}_i(x, s[i]) := \hat{P}_i(x)^{s[i]+}, \text{ for any } x \in \mathcal{X}[Z_i^P], s \in \mathcal{S}.$$

The new reward function is factored over $\tilde{\mathcal{X}} = \bigotimes_{i \in [l]} (\mathcal{X}[Z_i^P] \times \mathcal{S}_i)$, with reward functions to be

$$\tilde{R}_i(x, s[i]) = \hat{R}_i(x) + E_{R_i}(x),$$

for any $x \in \mathcal{X}[Z_i^R], s \in \mathcal{S}$.

**Claim 1.** *The factored set $\tilde{\mathcal{X}} = \mathcal{S} \times \tilde{\mathcal{A}}$ of the extended MDP $M_k$ satisfies each $|\tilde{\mathcal{X}}[Z_i^P]| \le LW$ for any $i \in [m]$ and each $|\tilde{\mathcal{X}}[Z_i^R]| \le LW$ for any $i \in [l]$.*

By Claim 1, any planner that efficiently solves the original MDP, can also solve the extended MDP. We find the best policy $\tilde{\pi}_k$ for $M_k$ using the planner. To run a policy $\pi_k$ on original action space, we choose $\pi_k$ such that $(s, \pi_k(s)) = f(s, \tilde{\pi}_k(s))$ for every $s \in \mathcal{S}$, where $f : \tilde{\mathcal{X}} \mapsto \mathcal{X}$ maps any new state-action pair to the pair it is extended from, i.e. $f((x, s)) = x$ for any $(x, s) \in \tilde{\mathcal{X}}$.

---

**Algorithm 1** PSRL and DORL

---

**Input:** $\mathcal{S}, \mathcal{A}$, accuracy $\rho$ for DORL and prior distribution for PSRL, $T$, encoding $\mathcal{G}$ and $L$, upper bound on the size of each factor set.
$k \leftarrow 1; t \leftarrow 1; t_k \leftarrow 1; T_k \leftarrow 1; \mathcal{H} = \{\}$.
**repeat**
    For DORL:
        Construct the extended MDP $M_k$ using error bounds:

$$E_{P_i}^k(s \mid x) = \min\{\sqrt{\frac{18\hat{P}_i(s|x)\log(c_{i,k})}{\max\{N_{P_i}^k(x), 1\}}} + \frac{18\log(c_{i,k})}{\max\{N_{P_i}^k(x), 1\}}, \hat{P}_i^k(s|x)\}, \qquad (2)$$

        for $c_{i,k} = 6mS_i|\mathcal{X}[Z_i^P]|t_k/\rho$ and

$$E_{R_i}^k = \sqrt{\frac{12\log(6l|\mathcal{X}[Z_i^R]|t_k/\rho)}{\max\{N_{R_i}(x), 1\}}}. \qquad (3)$$

        Compute $\tilde{\pi}_k = \pi(M_k)$ and find corresponding $\pi_k$ in original action space.
    For PSRL:
        Sample $M_k$ from $\phi(M|\mathcal{H})$.
        Compute $\pi_k = \pi(M_k)$.
    **for** $t = t_k$ **to** $t_k + T_k - 1$ **do**
        Apply action $a_t = \pi_k(s_t)$
        Observe new state $s_{t+1}$.
        Observe new rewards $r_{t+1} = (r_{t+1,1}, \ldots r_{t+1,l})$.
        $\mathcal{H} = \mathcal{H} \cup \{(s_t, a_t, r_{t+1}, s_{t+1})\}$.
    **end for**
    $k \leftarrow k + 1$.
    $T_k \leftarrow \lceil k/L \rceil; t_k \leftarrow t + 1$.
**until** $t_k > T$

---

## 4 Upper bounds for PSRL and DORL

We achieve the near-optimal Bayesian regret bound by PSRL and frequentist regret bound by DORL, respectively. Let $\tilde{O}$ denote the order ignoring the logarithmic term and the universal constant.

**Theorem 1** (Regret of PSRL). *Let $M$ be the factored MDP with graph structure $\mathcal{G} = \left(\{\mathcal{S}_i\}_{i=1}^m; \{\mathcal{X}_i\}_{i=1}^n; \{Z_i^R\}_{i=1}^l; \{Z_i^P\}_{i=1}^m\right)$, all $|\mathcal{X}[Z_i^R]|$ and $|\mathcal{X}[Z_j^P]| \leq L$, $|\mathcal{S}_i| \leq W$ and diameter upper bounded by $D$. Then if $\phi$ is the true prior distribution supported on the set of MDPs with diameter $\leq D$, then we bound Bayesian regret of PSRL:*

$$\mathbb{E}[R_T] = \tilde{O}(D(l + m\sqrt{W})\sqrt{TL}).$$

**Theorem 2** (Regret of DORL). *Let $M$ be the factored MDP with graph structure $\mathcal{G} = \left(\{\mathcal{S}_i\}_{i=1}^m; \{\mathcal{X}_i\}_{i=1}^n; \{Z_i^R\}_{i=1}^l; \{Z_i^P\}_{i=1}^m\right)$, all $|\mathcal{X}[Z_i^R]|$ and $|\mathcal{X}[Z_j^P]| \leq L$, $|\mathcal{S}_i| \leq W$ and diameter upper bounded by $D$. Then, with high probability, regret of DORL is upper bounded by:*

$$R_T = \tilde{O}(D(l + m\sqrt{W})\sqrt{TL}).$$

The two bounds match the frequentist regret bound in Jaksch et al. (2010) and Bayesian regret bound in Ouyang et al. (2017) for non-factored communicating MDP. We also give a condition of designing the speed of changing policies.

**Remark.** *Replacing the episode length in Algorithm 1 with any $\{T_k\}_{k=1}^K$ that satisfies $K = O(\sqrt{LT})$ and $T_k = O(\sqrt{T/L})$ for all $k \in [K]$, the frequentist bound in Theorem 2 still holds. Furthermore, if $\{T_k\}_{k=1}^K$ is fixed the Bayesian bound in Theorem 2 also holds.*

## 5  Lower Bound and Factored Span

Any regret bound depends on a difficulty measure determining the connectivity of the MDP. The upper bounds of DORL and PSRL use diameter. A tighter alternative is the span of bias vector (Bartlett and Tewari, 2009), defined as $sp(\boldsymbol{h}^*)$, where $\boldsymbol{h}^*$ is the bias vector of the optimal policy. However, none of those connectivity measures address the complexity of the graph structure. Indeed, some graph structure allows a tighter regret bound. In this section, we first show a lower bound with a Cartesian product structure. We further propose a new connectivity measure that can scale with the complexity of the graph structure.

**Large diameter case.** We consider a simple FMDP with infinite diameter but bounded span. The FMDP is a Cartesian product of two identical MDPs, $M_1$ and $M_2$ with $\mathcal{S} = \{0, 1, 2, 3\}$, $\mathcal{A} = \{1, 2\}$. The transition probability is chosen such that from any state and action pair, the next state will either move forward or move backward with probability one (state 0 is connected with state 3 as a circle).

We can achieve a low regret easily by learning each MDP independently. However, since the sum of the two states always keeps the same parity, vector state $(0, 1)$ can never be reached from $(0, 0)$. Thus, the FMDP has an infinite diameter. The span of bias vector, on the other hand, is upper bounded by $D(M_1) + D(M_2)$, which is tight in this case.

**Lower bound with only dependency on span.** Let us formally state the lower bound. Our lower bound casts some restrictions on the scope of transition probability, i.e. the scope contains itself, which we believe is a natural assumption. We provide Theorem 3 here.

**Theorem 3** (Lower bound). *For any algorithm, any graph structure satisfying $\mathcal{G} = \left(\{\mathcal{S}_i\}_{i=1}^n; \{\mathcal{S}_i \times \mathcal{A}_i\}_{i=1}^n; \{Z_i^R\}_{i=1}^n; \{Z_i^P\}_{i=1}^n\right)$ with $|\mathcal{S}_i| \leq W$, $|\mathcal{X}[Z_i^R]| \leq L$, $|\mathcal{X}[Z_i^P]| \leq L$ and $i \in Z_i^P$ for $i \in [n]$, there exists an FMDP with an optimal bias vector $\boldsymbol{h}$, such that for any initial state $s \in \mathcal{S}$, the expected regret of the algorithm after $T$ steps is $\Omega(\sqrt{sp(\boldsymbol{h})LT})$.*

The proof is given in Appendix B. Note that this bound of $\Omega(\sqrt{sp(\boldsymbol{h})LT})$ seems weaker than the lower bound $\sqrt{DSAT}$ when reduced to non-factored MDP case. However, the two bounds are exactly the same as in the construction from Jaksch et al. (2010), the diameter scales with span of bias vector by a constant factor. As we can see, the upper bound in Theorem 1 is larger than the lower bound by a factor of $\frac{D}{\sqrt{sp(\boldsymbol{h})}}$, $m$, $l$ and $\sqrt{W}$. We now discuss how to reduce the first three excesses.

### 5.1  Tighter connectivity measure

The mismatch in the dependence on $m$ is due to not taking the factor structure into account properly in the definition of the span. A tighter bound should be able to detect special structure, e.g. product of independent MDPs. For that purpose, recall that regret analysis involves in bounding a deviation term with the form

$$\sum_{s \in \mathcal{S}} \left(\tilde{P}^k(s) - P^k(s)\right) h_k(s) \leq span(\boldsymbol{h}_k) \max_s |\tilde{P}_k(s) - P_k(s)|_1,$$

where $\tilde{P}^k$ and $\tilde{P}^k$ are the transition probability of $M_k$ and $M$ using the policy $\tilde{\pi}_k$ and $\pi_k$. Here $h_k$ is the optimal bias vector of $M_k$. This inequality could be extremely loose as $\boldsymbol{h}_k$ has exponentially many dimensions. We use the following example to illustrate how to get a tighter inequality by looking at each factor on the bias vector.

**Reducing span by looking at each factor.** Let us write a bias vector of an FMDP with two factors as a matrix in the following table, where columns and rows represent the first and second dimension of the state vector respectively. The state set is given by $\{(x, y) : x \in \{x_1, x_2, x_3, x_4\}, y \in \{y_1, y_2, y_3\}\}$.

| factor 2 & factor 1 | $x_1$ | $x_2$ | $x_3$ | $x_4$ | span of rows |
|---|---|---|---|---|---|
| $y_1$ | 1 | 3 | 3 | 4 | 3 |
| $y_2$ | 4 | 3 | 6 | 7 | 4 |
| $y_3$ | 7 | 8 | 9 | 10 | 3 |
| span of columns | 6 | 5 | 6 | 6 | 9 |

While the total span of the vector is **9**, the span for each column and each row is at most **6** and **4**. We also notice that in the lower bound construction with two independent MDPs, the value that restricts the connectivity is the span of each column and each row. Even for the non-independent case, using the span of the whole vector will still lead a quite loose bound.

This example encourages us to construct a tighter connectivity measure *Factored Span* in Definition 5: for each factor, we write the bias vector into a matrix with the current factor as columns and the rest of factors as rows. Factored span is the maximum span of all the columns.

**Definition 5** (Factored span). *For an FMDP $M$ with an bias vector $\boldsymbol{h}$ of its optimal policy and a factorization of state space $\mathcal{S} = \bigotimes_{i=1}^{m} \mathcal{S}_i$, we define factored span $sp_1, \ldots, sp_m$ as:*

$$sp_i := \max_{s_{-i}} sp(\boldsymbol{h}(\cdot, s_{-i})) \text{ and let } Q(\boldsymbol{h}) := \sum_{i=1}^{m} sp_i,$$

*where $s_{-i} := (s_1, \ldots, s_{i-1}, s_{i+1}, \ldots, s_m)$ and $sp(h(\cdot, s_{-i})) := (h(s, s_{-i}))_{s \in \mathcal{S}_1}$.*

**Proposition 1.** *For any bias vector $\boldsymbol{h}$, $sp(\boldsymbol{h}) \leq Q(\boldsymbol{h}) \leq m\, sp(\boldsymbol{h})$. The first equality holds when the FMDP has the construction of Cartesian product of $m$ independent MDPs. The lower bound can also be written as $\Omega(\sqrt{Q(\boldsymbol{h})LT})$.*

## 5.2 Tighter upper bound

We now provide another algorithm called FSRL (Factored-span RL) with a tighter regret bound of $\tilde{O}(Q(\boldsymbol{h})\sqrt{WLT})$ as shown in Theorem 4. The bound reduces the gap on $m, l$ and replaces $D$ with the sum of factored span $Q$. Proposition 1 guarantees that $Q(\boldsymbol{h}) \leq msp(\boldsymbol{h}) \leq mD$ such that the upper bound is at least as good as the upper bound in Theorem 1.

FSRL (full description in Appendix D, Algorithm 2), mimics REGAL.C by solving the following optimization,

$$M = \arg\max_{M \in \mathcal{M}_k} \lambda^*(M) \quad \text{subject to} \quad Q(\boldsymbol{h}(M)) \leq Q \text{ for some prespecified } Q > 0,$$

where $\mathcal{M}_k$ is the confidence set defined in (2) and (3). FSRL relies on the computational oracle of optimizing average rewards over the confidence set with the sum of factored span bounded by a prespecified value. Therefore, FSRL cannot be run by just calling an FMDP planning oracle.

**Theorem 4** (Regret of FSRL (Factored-span RL)). *Let $M$ be the factored MDP with graph structure $\mathcal{G} = \left(\{\mathcal{S}_i\}_{i=1}^{m}; \{\mathcal{X}_i\}_{i=1}^{n}; \{Z_i^R\}_{i=1}^{l}; \{Z_i^P\}_{i=1}^{m}\right)$, all $|\mathcal{X}[Z_i^R]|$ and $|\mathcal{X}[Z_j^P]| \leq L$, $|\mathcal{S}_i| \leq W$, bias vector of optimal policy $\boldsymbol{h}$ and its sum of factored spans $Q(\boldsymbol{h})$. Then, with high probability, regret of FSRL is upper bounded by: $R_T = \tilde{O}(Q(\boldsymbol{h})\sqrt{WLT})$.*

Main idea is on bounding the deviation of transition probabilities between the true MDP and $M_k$ in episode $k$ with factored span. The details are shown in Appendix C.

## 6 Simulation

There are two previously available sample efficient and implementable algorithms for FMDPs: factored $E^3$ and factored Rmax (f-Rmax). F-Rmax was shown to have better empirical performance (Guestrin et al., 2002b). Thus, we compare PSRL, DORL and f-Rmax. For PSRL, at the start of each episode, we simply sample each factored transition probability and reward functions from a Dirichlet distribution and a Gaussian distribution, i.e. $P_i^k(x) \sim \text{Dirichlet}(N_{P_i}^t(\cdot, x)/c)$ and $R_i^k(x) \sim N(\hat{R}_i^k(x), c/N_{P_i}^t(x))$, where $c$ is searched over $(0.05, 0.1, 0.3, 0.75, 1, 5, 20)$. The total number of samplings for PSRL in each round is upper bounded by the number of parameters of the FMDP. For DORL, we replace the coefficients 18 and 12 in (2) and (3) with a hyper-parameter $c$ searched over

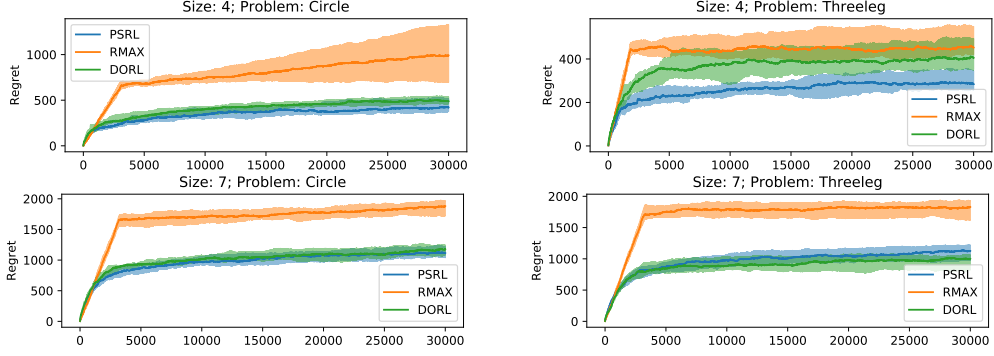

Figure 3: Regrets of PSRL, f-Rmax and DORL on circle and three-leg MDP with a size 4, 7. For PSRL, $c = 0.75$. For f-Rmax, $m = 300, 500, 500, 500$ and for DORL, $c = 0.03$ in Circle 4, Circle 7, Three-leg 4, Three-leg 7, respectively.

the set $\{0.05, 0.1, 0.3, 0.75, 1, 5, 20\}$. The choice of hyper-parameters is based on the fact that our regret analysis bounds the worst-case scenario, which lead to over-exploration, while in practice the level of exploration should be lower. For f-Rmax, $m$, the number of visits needed to be known are chosen from 100, 300, 500, 700 and the best choice is selected for each experiment.

For the approximate planner used by our algorithm, we implement approximate linear programming (Guestrin et al., 2003) with the basis $h_i(s) = s_i$ for $i \in [m]$. For regret evaluation, we use an accurate planner to find the true optimal average reward.

We compare three algorithms on computer network administrator domain with a circle and a three-leg structure (Guestrin et al., 2001; Schuurmans and Patrascu, 2002). To avoid the extreme case in our lower bound, both the MDPs are set to have limited diameters. The details on the environment are in Appendix E.

Figure 3 shows the regret of the two algorithms on circle and three-leg structure with a size 4, 7, respectively. Each experiment is run 20 times, with which median, 75% and 25% quantiles are computed. Our DORL and PSRL have very similar performance in all the environment except for Three-leg with a size 4. Optimal hyper-parameter for PSRL and DORL is stable in the way that $c$ around 0.75 and 0.03 are the optimal parameter for PSRL and DORL respectively for all the experiments. Note that we use the exact, not approximate, optimal reward in regret evaluation. So we see that DORL and PSRL was always able to find a near-optimal optimal policy despite the use of an approximate planner.

## 7   Discussion

FSRL relies on a harder computational oracle that is not efficiently solvable yet. Fruit et al. (2018) achieved a regret bound depending on span using an implementable Bias-Span-Constrained value iteration on non-factored MDP. It remains unknown whether FSRL could be approximately solved using an efficient implementation.

In non-factored MDP, Zhang and Ji (2019) achieved the lower bound of $\sqrt{DSAT}$. On the lower bound of FMDP, it remains an open problem to close the remaining gap involving $\sqrt{W}$ and $\sqrt{Q}$. Recently, Sadegh Talebi et al. (2020) improved the exploration using a Berstein-style confidence set and was able to reduce the extra term on $\sqrt{W}$. However, their algorithm depends on the harder planning oracle on bounded FMDP. UCRL3 (Bourel et al., 2020) improves the dependence on actions and states by restricting the set of successor states of the state-action pair on non-factored MDP. We believe this construction is quite independent from the FMDP construction and the similar adaptation can easily be applied. Our algorithms require the full knowledge of the graph structure of the FMDP, which can be impractical. The structural learning scenario has been studied by Strehl et al. (2007); Chakraborty and Stone (2011); Hallak et al. (2015). Their algorithms either rely on an admissible structure learner or do not have a regret or sample complexity guarantee. Recently, Rosenberg and Mansour (2020) proposed an oracle-efficient structure learner with regret guarantee.

## 8 Broader Impacts

As a theoretical paper, we can not foresee any direct societal consequences in the near future. Factored MDP, the main problem we study in this paper, may be used in multi-agent Reinforcement Learning scenario.

**Acknowledgements.** We acknowledge the support of NSF via grant IIS-2007055.

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
