[Supplementary Material]

# A Proof of Theorem 1 and 2

A standard regret analysis consists of proving the optimism, bounding the deviations and bounding the probability of failing the confidence set. Our analysis follows the standard procedure while adapting them to a FMDP setting.

**Some notations.** For simplicity, we let $\pi^*$ denote the optimal policy of the true MDP, $\pi(M)$. Let $t_k$ be the starting time of episode $k$ and $K$ be the total number of episodes. Since $\tilde{R}^k(x, s)$ for any $(x, s) \in \tilde{\mathcal{X}}$ does not depend on $s$, we also let $\tilde{R}^k(x)$ denote $\tilde{R}^k(x, s)$ for any $s$. Let $\lambda^*$ and $\lambda_k$ denote the optimal average reward for $M$ and $M_k$.

**Confidence set.** Before proving the theorems, we first introduce the confidence set for both transition probability and reward functions. Let $\mathcal{M}_k$ be the confidence set of FMDPs at the start of episode $k$ with the same factorization, such that for and each $i \in [l]$,

$$|R_i(x) - \hat{R}_i^k(x)| \le E_{R_i}^k(x), \forall x \in \mathcal{X}[Z_i^R],$$

where $E_{R_i}^k(x) := \sqrt{\frac{12 \log(6l|\mathcal{X}[Z_i^R]|t_k/\rho)}{\max\{N_{R_i}^k(x), 1\}}}$ as defined in (3);

and for each $j \in [m]$

$$|P_j(s|x) - \hat{P}_j^k(s|x)| \le E_{P_j}^k(s|x), \forall x \in \mathcal{X}[Z_j^P], s \in \mathcal{S}_j,$$

where $W_{P_j}^k(s|x)$ is defined in (2). It can be shown that

$$|P_j(x) - \hat{P}_j^k(x)|_1 \le 2\sqrt{\frac{18|\mathcal{S}_i| \log(6S_i m|\mathcal{X}[Z_i^P]|t_k/\rho)}{\max\{N_{P_i}^k(x), 1\}}},$$

where $\bar{E}_{P_i}^k(x) := 2\sqrt{\frac{18|\mathcal{S}_i| \log(6S_i m|\mathcal{X}[Z_i^P]|t_k/\rho)}{\max\{N_{P_i}^k(x), 1\}}}$.

In the following analysis, we all assume that the true MDP $M$ for both PSRL and DORL are in $\mathcal{M}_k$ and $M_k$ by PSRL are in $\mathcal{M}_k$ for all $k \in [K]$. In the end, we will bound the regret caused by the failure of confidence set.

**Regret decomposition.** We follow the standard regret analysis framework by Jaksch et al. (2010). We first decompose the total regret into three parts in each episode:

$$R_T = \sum_{t=1}^T (\lambda^* - r_t)$$

$$= \sum_{k=1}^K \sum_{t=t_k}^{t_{k+1}-1} (\lambda^* - \lambda_k) \tag{4}$$

$$+ \sum_{k=1}^K \sum_{t=t_k}^{t_{k+1}-1} (\lambda_k - R(s_t, a_t)) \tag{5}$$

$$+ \sum_{k=1}^K \sum_{t=t_k}^{t_{k+1}-1} (R(s_t, a_t) - r_t). \tag{6}$$

Using Hoeffding's inequality, the regret caused by (6) can be upper bounded by $\sqrt{\frac{5}{2}T \log\left(\frac{8}{\rho}\right)}$, with probability at least $\frac{\rho}{12}$.

## A.1 Bounding term (4)

We bound the regret caused by (4).

**PSRL.** For PSRL, since we use fixed episodes, we follow the techniques from Osband et al. (2013) and show that the expectation of (4) equals to zero.

**Lemma 1** (Lemma 1 in Osband et al. (2013)). *If $\phi$ is the distribution of $M$, then, for any $\sigma(H_{t_k}) -$ measurable function $g$,*

$$\mathbb{E}\left[g\left(M\right)|H_{t_k}\right] = \mathbb{E}\left[g\left(M_k\right)|H_{t_k}\right].$$

We let $g = \lambda(M, \pi(M))$. As $g$ is a $\sigma(H_{t_k}) - measurable$ function. Since $t_k$, $K$ are fixed value for each $k$, we have $(4) = \mathbb{E}[\sum_{k=1}^{K}\sum_{t=t_k}^{t_{k+1}-1}(\lambda^* - \lambda_k)] = 0$.

**DORL.** For DORL, we need to prove optimism, i.e, $\lambda(M_k, \tilde{\pi}_k) \geq \lambda^*$ with high probability. We follow the proof in Agrawal and Jia (2017). In the case of FMDP, we show that for any policy $\pi$ for the true FMDP, there exists a policy $\tilde{\pi}$ for $M_k$ such that $(P(M_k, \tilde{\pi}) - P(M, \pi))\boldsymbol{h} \geq 0$ for any $h \in \mathbb{R}^S$. This is proved in Lemma 2.

**Lemma 2.** *For any policy $\pi$ for $M$ and any vector $\boldsymbol{h} \in \mathbb{R}^S$, let $\tilde{\pi}$ be the policy for $M_k$ satisfying $\tilde{\pi}(s) = (\pi(s), s^*)$, where $s^* = \arg\max_s \boldsymbol{h}(s)$. Then, given $M \in \mathcal{M}_k$, $(P(M_k, \tilde{\pi}) - P(M, \pi))\boldsymbol{h} \geq 0$.*

*Proof.* We fix some $s \in \mathcal{S}$ and let $x = (s, \pi(s)) \in \mathcal{X}$. Recall that for any $s_i \in \mathcal{S}_i$, $\Delta_i^k(s_i|x) =$

$$\min\left\{\sqrt{\frac{18\hat{P}_i^k(s_i|x)\log(c_{i,k})}{\max\left\{N_{P_i}^k(x),1\right\}}} + \frac{18\log(c_{i,k})}{\max\left\{N_{P_i}^k(x),1\right\}}, \hat{P}_i^k(s_i|x)\right\}.$$

and define $P_i^-(\cdot|x) = \hat{P}_i^k(\cdot|x) - \Delta_i^k(\cdot|x)$. Slightly abusing the notations, let $\tilde{\boldsymbol{P}} = P(M_k, \tilde{\pi})_{s,\cdot}$, $\boldsymbol{P} = P(M, \pi)_{s,\cdot}$. Define two $S$-dimensional vectors $\hat{\boldsymbol{P}}$ and $\boldsymbol{P}^-$ with $\hat{\boldsymbol{P}}(\bar{s}) = \prod_i \hat{P}_i(\bar{s}[Z_i^P]|x)$ and $\boldsymbol{P}^-(\bar{s}) = \Pi_i P_i^-(\bar{s}[Z_i^P]|x)$ for $\bar{s} \in \mathcal{S}$.

As $M \in \mathcal{M}_k$, $\boldsymbol{P}^- \leq \boldsymbol{P}$. Define $\boldsymbol{\alpha} := \hat{\boldsymbol{P}} - \boldsymbol{P} \leq \hat{\boldsymbol{P}} - \boldsymbol{P}^- =: \boldsymbol{\Delta}$. Without loss of generality, we let $\max_s \boldsymbol{h}(s) = D$.

$$\begin{aligned}
\sum_i \tilde{\boldsymbol{P}}(i)\boldsymbol{h}(i) &= \sum_i \boldsymbol{P}(i)^-\boldsymbol{h}(i) + D\left(1 - \sum_j \boldsymbol{P}(j)^-\right) \\
&= \sum_i \boldsymbol{P}(i)^-\boldsymbol{h}(i) + D\sum_j \boldsymbol{\Delta}(j) \\
&= \sum_i \left(\hat{\boldsymbol{P}}(i) - \boldsymbol{\Delta}(i)\right)\boldsymbol{h}(i) + D\boldsymbol{\Delta}(i) \\
&= \sum_i \hat{\boldsymbol{P}}(i)\boldsymbol{h}(i) + (D - \boldsymbol{h}(i))\boldsymbol{\Delta}(i) \\
&\geq \sum_i \hat{\boldsymbol{P}}(i)\boldsymbol{h}(i) + (D - \boldsymbol{h}(i))\boldsymbol{\alpha}(i) \\
&= \sum_i \left(\hat{\boldsymbol{P}}(i) - \boldsymbol{\alpha}(i)\right)\boldsymbol{h}(i) + D\boldsymbol{\alpha}(i) \\
&= \sum_i \boldsymbol{P}(i)\boldsymbol{h}(i) + D\sum_i \boldsymbol{\alpha}(i) = \sum_i \boldsymbol{P}(i)\boldsymbol{h}(i)
\end{aligned}$$

$\square$

**Corollary 1.** *Let $\tilde{\pi}^*$ be the policy that satisfies $\tilde{\pi}^*(s) = (\pi^*(s), s^*)$, where $s^* = \arg\max_s \boldsymbol{h}(M)_s$ and $\pi^*$ is the true optimal policy for $M$. Then $\lambda(M_k, \tilde{\pi}^*, s_1) \geq \lambda^*$ for any starting state $s_1$.*

*Proof.* Let $\boldsymbol{d}(s_1) := \boldsymbol{d}(M_k, \tilde{\pi}^*, s_1) \in \mathbb{R}^{1 \times S}$ be the row vector of stationary distribution starting from some $s_1 \in \mathcal{S}$. By optimal equation,

$$
\begin{aligned}
&\lambda(M_k, \tilde{\pi}^*, s_1) - \lambda^* \\
&= \boldsymbol{d}(s_1)\boldsymbol{R}(M_k, \tilde{\pi}^*) - \lambda^*(\boldsymbol{d}(s_1)\mathbf{1}) \\
&= \boldsymbol{d}(s_1)(\boldsymbol{R}(M_k, \tilde{\pi}^*) - \lambda^*\mathbf{1}) \\
&= \boldsymbol{d}(s_1)(\boldsymbol{R}(M_k, \tilde{\pi}^*) - \boldsymbol{R}(M, \pi^*)) \\
&\quad + \boldsymbol{d}(s_1)(I - P(M, \pi^*))\boldsymbol{h}(M) \\
&\geq \boldsymbol{d}(s_1)(\boldsymbol{R}(M_k, \tilde{\pi}^*) - \boldsymbol{R}(M, \pi^*)) \\
&\quad + \boldsymbol{d}(s_1)(P(M_k, \tilde{\pi}^*) - P(M, \pi^*))\boldsymbol{h}(M) \\
&\geq 0,
\end{aligned}
$$

where the last inequality is by Lemma 2 and Corollary 1 follows. $\qquad\square$

Thereon, $\lambda(M_k, \tilde{\pi}_k) \geq \lambda(M_k, \tilde{\pi}^*, s_1) \geq \lambda^*$. The total regret of (4) $\leq 0$.

## A.2 Regret caused by deviation (5)

We further bound regret caused by (5), which can be decomposed into the deviation between our brief $M_k$ and the true MDP. We first show that the diameter of $M_k$ can be upper bounded by $D$.

**Bounded diameter.** We need diameter of extended MDP to be upper bounded to give a sublinear regret. For PSRL, since prior distribution has no mass on MDP with diameter greater than $D$, the diameter of MDP from posterior is upper bounded by $D$ almost surely. For DORL, we have the following Lemma 3.

**Lemma 3.** *When $M$ is in the confidence set $\mathcal{M}_k$, the diameter of the extended MDP $D(M_k) \leq D$.*

*Proof.* Fix a $s_1 \neq s_2$, there exist a policy $\pi$ for $M$ such that the expected time to reach $s_2$ from $s_1$ is at most $D$, without loss of generality we assume $s_2$ is the last state. Let $E$ be the $(S-1) \times 1$ vector with each element to be the expected time to reach $s_2$ except for itself. We find $\tilde{\pi}$ for $M_k$ such that the expected time to reach $s_2$ from $s_1$ can be bounded by $D$. We choose the $\tilde{\pi}$ that satisfies $\tilde{\pi}(s) = (\pi(s), s_2)$.

Let $Q$ be the transition matrix under $\tilde{\pi}$ for $M_k$. Let $Q^-$ be the matrix removing $s_2$-th row and column and $P^-$ defined in the same way for $M$. We immediately have $P^{-1}E \geq Q^{-1}E$, given $M \in \mathcal{M}_k$. Let $\tilde{E}$ be the expected time to reach $s_2$ from every other states except for itself under $\tilde{\pi}$ for $M_k$.

We have $\tilde{E} = \mathbf{1} + Q^-\tilde{E}$. The equation for $E$ gives us $E = \mathbf{1} + P^-E \geq \mathbf{1} + Q^-E$. Therefore,

$$
\tilde{E} = (1 - Q^-)^{-1}\mathbf{1} \leq E,
$$

and $\tilde{E}_{s_1} \leq E_{s_1} \leq D$. Thus, $D(M_k) \leq D$. $\qquad\square$

**Deviation bound.** Now we formally bound (5). In this section, the regrets for PSRL and DORL can be bounded in the same way. Let $\nu_k(s, a)$ be the number of visits on $s, a$ in episode $k$ and $\boldsymbol{\nu}_k$ be the row vector of $\nu_k(\cdot, \pi_k(\cdot))$. Let $\Delta_k = \sum_{s,a} \nu_k(s, a)(\lambda(M_k, \tilde{\pi}_k) - R(s, a))$. Using optimal equation,

$$
\begin{aligned}
\Delta_k &= \sum_{s,a} \nu_k(s, a)\left[\lambda(M_k, \tilde{\pi}_k) - \tilde{R}^k(s, a)\right] \\
&\quad + \sum_{s,a} \nu_k(s, a)\left[\tilde{R}^k(s, a) - R(s, a)\right] \\
&= \boldsymbol{\nu}_k(\tilde{P}^k - I)\boldsymbol{h}_k + \boldsymbol{\nu}_k(\tilde{\boldsymbol{R}}^k - \boldsymbol{R}^k) \\
&= \underbrace{\boldsymbol{\nu}_k(P^k - I)\boldsymbol{h}_k}_{①} + \underbrace{\boldsymbol{\nu}_k(\tilde{P}^k - P^k)\boldsymbol{h}_k}_{②} + \underbrace{\boldsymbol{\nu}_k(\tilde{\boldsymbol{R}}^k - \boldsymbol{R}^k)}_{③},
\end{aligned}
$$

where $\tilde{P}^k := P(M_k, \tilde{\pi}_k), P^k := P(M, \pi_k), \boldsymbol{h}_k := \boldsymbol{h}^*(M_k)$, and $\tilde{\boldsymbol{R}}^k := \boldsymbol{R}(M_k, \tilde{\pi}_k), \boldsymbol{R}^k := \boldsymbol{R}(M, \pi_k)$.

Using Azuma-Hoeffding inequality and the same analysis in Jaksch et al. (2010), we bound ① with probability at least $1 - \frac{\rho}{12}$,

$$\sum_k ① = \sum_k \boldsymbol{\nu}_k \left( P^k - I \right) \boldsymbol{h}_k \leq D\sqrt{\frac{5}{2}T \log\left(\frac{8}{\rho}\right)} + KD. \tag{7}$$

To bound ② and ③, we analyze the deviation in transition and reward function between $M$ and $M_k$. For DORL, the deviation in transition probability is upper bounded by

$$\max_{s'} |\tilde{P}_i^k(x, s') - \hat{P}_i^k(x)|_1$$

$$\leq \min\{2 \sum_{s \in \mathcal{S}_i} E_{P_i}^k(s \mid x), 1\}$$

$$\leq \min\{2\bar{E}_{P_i}^k(x), 1\} \leq 2\bar{E}_{P_i}^k(x),$$

The deviation in reward function $|\tilde{R}_i^k - \hat{R}_i^k|(x) \leq E_{R_i}^k(x)$.

For PSRL, since $M_k \in \mathcal{M}_k$, $|\tilde{P}_i^k - \hat{P}_i^k|(x) \leq \bar{E}_{P_i}^k(x)$ and $|\tilde{R}_i^k - \hat{R}_i^k|(x) \leq E_{R_i}^k(x)$.

Decomposing the bound for each scope provided by $M \in \mathcal{M}_k$ and $M_k$ for PSRL $\in \mathcal{M}_k$, it holds for both PSRL and DORL that:

$$\sum_k ② \leq 3 \sum_k D \sum_{i=1}^{m} \sum_{x \in \mathcal{X}[Z_i^P]} \nu_k(x) \bar{E}_{P_i}^k(x), \tag{8}$$

$$\sum_k ③ \leq 2 \sum_k \sum_{i=1}^{l} \sum_{x \in \mathcal{X}[Z_i^R]} \nu_k(x) E_{R_i}^k(x); \tag{9}$$

where with some abuse of notations, define $\nu_k(x) = \sum_{x' \in \mathcal{X}: x'[Z_i] = x} \nu_k(x')$ for $x \in \mathcal{X}[Z_i]$. The second inequality is from the fact that $|\tilde{P}^k(\cdot|x) - P^k(\cdot|x)|_1 \leq \sum_1^m |\tilde{P}_i^k(\cdot|x[Z_i^R]) - P_i^k(\cdot|x[Z_i^R])|_1$ (Osband and Van Roy, 2014).

## A.3 Bound (7), (8) and (9) by balancing episode length and episode number

We give a general criterion for bounding (7), (8) and (9), which we believe, is a new technique. We first introduce Lemma 4 which implies that bounding (7), (8) and (9) is to balance total number of episodes and the length of the longest episode. The proof, relies on defining the last episode $k_0$, such that $N_{k_0}(x) \leq \nu_{k_0}(x)$.

**Lemma 4.** *For any fixed episodes $\{T_k\}_{k=1}^K$, if there exists an upper bound $\bar{T}$, such that $T_k \leq \bar{T}$ for all $k \in [K]$, we have the bound*

$$\sum_{x \in \mathcal{X}[Z]} \sum_k \nu_k(x)/\sqrt{\max\{1, N_k(x)\}} \leq L\bar{T} + \sqrt{LT},$$

*where $Z$ is any scope with $|\mathcal{X}[Z]| \leq L$, and $\nu_k(x)$ and $N_k(x)$ are the number of visits to $x$ in and before episode $k$. Furthermore, total regret of (7), (8) and (9) can be bounded by $\tilde{O}\left((\sqrt{W}Dm + l)(L\bar{T} + \sqrt{LT}) + KD\right)$*

*Proof.* We bound the random variable $\sum_{k=1}^K \frac{\nu_k(x)}{\sqrt{\max\{N_k(x),1\}}}$ for every $x \in \mathcal{X}[Z]$, where $\nu_k(x) = \sum_{t=t_k}^{t_{k+1}-1} \mathbb{1}(x_t = x)$ and $N_k(x) = \sum_{i=1}^{k-1} \nu_i(x)$.

Let $k_0(x)$ be the largest $k$ such that $N_k(x) \leq \nu_k(x)$. Thus $\forall k \geq k_0(x), N_k(x) > \nu_k(x)$, which gives $N_t(x) := N_k(x) + \sum_{\tau=t_k}^t \mathbb{1}(x_\tau = x) < 2N_k(x)$ for $t_k \leq t < t_{k+1}$.

Conditioning on $k_0(x)$, we have

$$\sum_{k=1}^{K} \frac{\nu_k(x)}{\sqrt{\max\{N_k(x),1\}}}$$

$$\leq N_{k_0(x)}(x) + \nu_{k_0(x)}(x) + \sum_{k>k_0(x)} \frac{\nu_k(x)}{\sqrt{\max\{N_k(x),1\}}}$$

$$\leq 2\nu_{k_0(x)}(x) + \sum_{k>k_0(x)} \frac{\nu_k(x)}{\sqrt{\max\{N_k(x),1\}}}$$

$$\leq 2\bar{T} + \sum_{k>k_0(x)} \frac{\nu_k(x)}{\sqrt{\max\{N_k(x),1\}}},$$

where the first inequality uses $\max\{N_k(x),1\} \geq 1$ for $k = 1, \ldots k_0(x)$, the second inequality is by the fact that $N_{k_0(x)}(x) \leq \nu_{k_0(x)}(x)$ and the third one is by $\nu_{k_0}(x) \leq T_{k_0(x)} \leq T_K$.

And letting $k_1(x) = k_0(x) + 1$ and $N(x) := N_K(x) + \nu_K(x)$, we have

$$\sum_{k>k_0(x)} \frac{\nu_k(x)}{\sqrt{\max\{N_k(x),1\}}}$$

$$\leq \sum_{t=t_{k_1(x)}}^{T} 2\frac{\mathbb{1}(x_t = x)}{\sqrt{\max\{N_t(x),1\}}}$$

$$\leq \sum_{t=t_{k_1(x)}}^{T} 2\frac{\mathbb{1}(x_t = x)}{\sqrt{\max\{N_t(x) - N_{k_1(x)},1\}}}$$

$$\leq 2\int_{1}^{N(x)-N_{k_1(x)}} \frac{1}{\sqrt{x}}dx$$

$$\leq (2+\sqrt{2})\sqrt{N(x)}.$$

Given any $k_0(x)$, we can bound the term with a fixed value $2\bar{T} + (2+\sqrt{2})\sqrt{N(x)}$. Thus, the random variable $\sum_{k=1}^{K} \frac{\nu_k(x)}{\sqrt{\max\{N_k(x),1\}}}$ is upper bounded by $2\bar{T} + (2+\sqrt{2})\sqrt{N(x)}$ almost surely. Finally, $\sum_x \sum_{k=1}^{K} \frac{\nu_k(x)}{\sqrt{\max\{N_k(x),1\}}} \leq L\bar{T} + (2+\sqrt{2})\sqrt{LT}$. The regret by (8) is

$$\sum_k 3D \sum_{i\in[m]} \sum_{x\in\mathcal{X}[Z_i^P]} \nu_k(x)\bar{W}_{P_i}^k(x)$$

$$= \tilde{O}(\sqrt{W}Dm(L\bar{T} + \sqrt{LT}) + KD).$$

The regret by (9) is

$$\sum_k 2 \sum_{i\in[l]} \sum_{x\in\mathcal{X}[Z_i^R]} \nu_k(x)\bar{W}_{R_i}^k(x) = \tilde{O}(l(L\bar{T} + \sqrt{LT}) + KD).$$

The last statement is completed by directly summing (7), (8) and (9). $\qquad\square$

Instead of using the doubling trick that was used in Jaksch et al. (2010). We use an arithmetic progression: $T_k = \lceil k/L \rceil$ for $k \geq 1$. As in our algorithm, $T \geq \sum_{k=1}^{K-1} T_k \geq \sum_{k=1}^{K-1} k/L = \frac{(K-1)K}{2L}$, we have $K \leq \sqrt{3LT}$ and $T_k \leq T_K \leq K/L \leq \sqrt{3T/L}$ for all $k \in [K]$. Thus, by Lemma 4, putting (6), (7), (9), (8) together, the total regret for $M \in \mathcal{M}_k$ is upper bounded by

$$\tilde{O}\big((\sqrt{W}Dm + l)\sqrt{LT}\big), \qquad (10)$$

with a probability at least $1 - \frac{\rho}{6}$.

### A.4 Failure of the confidence set

For the failure of confidence set, we prove the following Lemma.

**Lemma 5.** *For all $k \in [K]$, with probability greater than $1 - \frac{3\rho}{8}$, $M \in \mathcal{M}_k$ holds.*

*Proof.* We first deal with the probabilities, with which in each round a reward function of the true MDP $M$ is not in the confidence set. Using Hoeffding's inequality, we have for any $t, i$ and $x \in \mathcal{X}[Z_i^R]$,

$$
\mathbb{P} \left\{ |\hat{R}_i^t(x) - R_i(x)| \geq \sqrt{\frac{12 \log(6l|\mathcal{X}[Z_i^R]|t/\rho)}{\max\{1, N_{R_i}^t(x)\}}} \right\}
$$

$$
\leq \frac{\rho}{3l|\mathcal{X}[Z_i^R]|t^6}, \text{ with a summation } \leq \frac{3}{12}\rho.
$$

Thus, with probability at least $1 - \frac{3\rho}{12}$, the true reward function is in the confidence set for every $t \leq T$.

For the transition probability, we use a different concentration inequality.

**Lemma 6** (Multiplicative Chernoff Bound (Kleinberg et al., 2008) Lemma 4.9). *Consider $n$, i.i.d random variables $X_1, \ldots, X_n$ on $[0, 1]$. Let $\mu$ be their mean and let $X$ be their average. Then with probability $1 - \rho$,*

$$
|X - \mu| \leq \sqrt{\frac{3 \log(2/\rho)X}{n}} + \frac{3 \log(2/\rho)}{n}.
$$

Using Lemma 6, for each $x, i, k$, it holds that with probability $1 - \rho/(6m \left|\mathcal{X}\left[Z_i^P\right]\right| t_k^6)$,

$$
|\hat{P}_i(\cdot|x) - P_i(\cdot|x)|_1 \leq \sqrt{\frac{18 S_i \log(c_{i,k})}{\max\{N_{P_i}^k(x), 1\}}} + \frac{18 \log(c_{i,k})}{\max\{N_{P_i}^k(x), 1\}}.
$$

Then with a probability $1 - \frac{3\rho}{24}$, it holds for all $x, i, k$. Therefore, with a probability $1 - \frac{3\rho}{8}$, the true MDP is in the confidence set for each $k$. $\qquad\square$

Combined with (10), with probability at least $1 - \frac{2\rho}{3}$ the regret bound in Theorem 2 holds.

For PSRL, $M_k$ and $M$ has the same posterior distribution. The expectation of the regret caused by $M \notin \mathcal{M}_k$ and $M_k \notin \mathcal{M}_k$ are the same. Choosing sufficiently small $\rho \leq \sqrt{1/T}$, Theorem 1 follows.

## B  Proof of the lower bound

Our lower bound construction is a Cartesian product of $n$ independent MDPs. We start by discussing the bias vector of such FMDP in Lemma 7.

**Lemma 7.** *Let $M^+$ be the Cartesian product of $n$ independent MDPs $\{M_i\}_{i=1}^n$, each with a span of bias vector $sp(h_i)$. The optimal policy for $M^+$ has a span $sp(h^+) = \sum_i sp(h_i)$.*

*Proof.* Let $\lambda_i^*$ for $i \in [n]$ be the optimal gain of each MDP. Optimal gain of $M^+$ is direct $\lambda^* = \sum_{i \in [n]} \lambda_i^*$. As noted in Puterman (2014) (8.2.3), by the definition of bias vector we have

$$
h_i(s) = \mathbb{E}[\sum_{t=1}^{\infty} (r_t^i - \lambda_i^*) \mid s_1^i = s], \quad \forall s \in \mathcal{S}_i,
$$

where $r_t^i$ is the reward of the $i$-th MDP at time $t$ and $s_t^i := s_t[i]$.

The lemma is directly by

$$h^+(s) = \mathbb{E}[\sum_{t=1}^{\infty}(r_t - \lambda^*) \mid s_1 = s]$$

$$= \mathbb{E}[\sum_{t=1}^{\infty}(\sum_{i \in [n]}(r_t^i - \lambda_i^*)) \mid s_1 = s]$$

$$= \sum_{i \in [n]} \mathbb{E}[\sum_{t=1}^{\infty}(r_t^i - \lambda_i^*) \mid s_1^i = s[i]]$$

$$= \sum_{i \in [n]} h_i(s[i]).$$

We immediately have $sp(h^+) = \sum_i sp(h_i)$. □

Recall Theorem 3 states *for any algorithm, any graph structure satisfying* $\mathcal{G} = \left(\{\mathcal{S}_i\}_{i=1}^n ; \{\mathcal{S}_i \times \mathcal{A}_i\}_{i=1}^n ; \{Z_i^R\}_{i=1}^n ; \{Z_i^P\}_{i=1}^n\right)$ *with* $|\mathcal{S}_i| \leq W$, $|\mathcal{X}[Z_i^R]| \leq L$, $|\mathcal{X}[Z_i^P]| \leq L$ *and* $i \in Z_i^P$ *for* $i \in [n]$*, there exists an FMDP with an optimal bias vector* $\boldsymbol{h}^+$*, such that for any initial state* $s \in \mathcal{S}$*, the expected regret of the algorithm after* $T$ *step is*

$$\Omega(\sqrt{sp(\boldsymbol{h}^+)LT}). \tag{11}$$

*Proof.* Let $l = |\cup_i^n Z_i^R|$. As $i \in Z_i^P$, a special case is the FMDP with graph structure $\mathcal{G} = \left(\{\mathcal{S}_i\}_{i=1}^n ; \{\mathcal{S}_i \times \mathcal{A}_i\}_{i=1}^n ; \{\{i\}\}_{i=1}^l \text{ and } \{\emptyset\}_{i=l+1}^n ; \{\{i\}\}_{i=1}^n\right)$, which can be decomposed into $n$ independent MDPs as in the previous example. Among the $n$ MDPs, the last $n-l$ MDPs are trivial. By simply setting the rest $l$ MDPs to be the construction used by Jaksch et al. (2010), which we refer to as "JAO MDP", the regret for each MDP with the span $sp(\boldsymbol{h})$, is $\Omega(\sqrt{sp(\boldsymbol{h})WT})$ for $i \in [l]$. The total regret is $\Omega(l\sqrt{sp(\boldsymbol{h})WT})$.

Using Lemma 7, $sp(\boldsymbol{h}^+) = l\,sp(\boldsymbol{h})$ and the total expected regret is $\Omega(\sqrt{l\,sp(\boldsymbol{h}^+)WT})$. Normalizing the reward function to be in $[0,1]$, the expected regret of the FMDP is $\Omega(\sqrt{sp(\boldsymbol{h}^+)WT})$, which completes the proof. □

## C  Proof of Theorem 4

The only difference between the proof of Theorem 4 and 2 lies in the bound of term ②.

*Proof.* Starting from ②, for each $s \in \mathcal{S}$, we bound $(\tilde{P}^k(\cdot \mid s) - P^k(\cdot \mid s))\boldsymbol{h}_k$. For simplicity, we remove the subscriptions of $s$ and use $\tilde{P}^k$ and $P^k$ to denote the vector for $s$-th row of the two matrix.

$$\sum_{s \in \mathcal{S}} (\tilde{P}^k(s) - P^k(s)) h_k(s)$$

$$= \sum_{s_1 \in \mathcal{S}_1} \sum_{s_{-1} \in \mathcal{S}^{-1}} (P_1(s_1)P_{-1}(s_{-1}) - \tilde{P}_1(s_1)\tilde{P}_{-1}(s_{-1})) h_k(s_1, s_{-1})$$

$$= \sum_{s_1} \left[ (P_1(s_1) - \tilde{P}_1(s_1)) \sum_{s_{-1}} \tilde{P}_{-1}(s_{-1}) h_k(s_1, s_{-1}) \right] +$$

$$\sum_{s_{-1}} \left[ (P_{-1}(s_{-1}) - \tilde{P}_{-1}(s_{-1})) \sum_s P_1(s_1) h_k(s_1, s_{-1}) \right]$$

$$= \sum_{s_1} (P_1(s_1) - \tilde{P}_1(s_1)) h_{1k}(s_1) + \sum_{s_{-1}} (P_{-1}(s_{-1}) - \tilde{P}_{-1}(s_{-1})) h_{-1k}(s_{-1}),$$

where $h_{1k}(s_1) := \sum_{s_{-1}} \tilde{P}_{-1}(s_{-1}) h_k(s_1, s_{-1})$ and $h_{-1k}(s_{-1}) := \sum_{s_1} P_1(s_1) h_k(s_1, s_{-1})$. As $span(h_{1k}) \le sp_1(M_k)$,

$$\sum_{s \in \mathcal{S}} (\tilde{P}^k(s) - P^k(s)) h_k(s) \le |P_1 - \tilde{P}_1|_1 sp_1(M_k) + \sum_{s_{-1}} (P_{-1}(s_{-1}) - \tilde{P}_{-1}(s_{-1})) h_{-1k}(s_{-1}). \quad (12)$$

By applying (12) recurrently, we have

$$\sum_{s \in \mathcal{S}} (\tilde{P}^k(s) - P^k(s)) h_k(s) \le \sum_{i=1}^{m} |P_i - \tilde{P}_i|_1 sp_i(M_k).$$

Note that $sp_i(M_k)$ is generally smaller than $span(h_k)$. In our lower bound case each $sp_i = \frac{1}{m} span(h_k)$, which improves our upper bound by a scale of $1/m$.

The reduction of $l$ can be achieved by bounding each factored reward to be in $[1, 1/l]$. The following proof remains the same. $\qquad \square$

## D   FSRL algorithm

Here we provide a complete description of the FSRL algorithm that was omitted in the main paper due to space considerations.

---

**Algorithm 2** FSRL

---

    **Input:** $\mathcal{S}, \mathcal{A}, T$, encoding $\mathcal{G}$ and upper bound on sum of factored span $Q$.
    $k \leftarrow 1; t \leftarrow 1; t_k \leftarrow 1; T_k = 1; \mathcal{H} \leftarrow \{\}$
    **repeat**
        Choose $M_k \in \mathcal{M}_k$ by solving the following optimization over $M \in \mathcal{M}_k$,

$$\max \lambda^*(M) \quad \text{subject to} \quad Q(h) \le Q \text{ for } h \text{ being the bias vector of } M.$$

        Compute $\tilde{\pi}_k = \pi(M_k)$.
        **for** $t = t_k$ **to** $t_k + T_k - 1$ **do**
            Apply action $a_t = \pi_k(s_t)$
            Observe new state $s_{t+1}$
            Observe new rewards $r_{t+1} = (r_{t+1,1}, \dots r_{t+1,l})$
            $\mathcal{H} = \mathcal{H} \cup \{(s_t, a_t, r_{t+1}, s_{t+1})\}$
            $t \leftarrow t + 1$
        **end for**
        $k \leftarrow k + 1.$
        $T_k \leftarrow \lceil k/L \rceil; t_k \leftarrow t + 1.$
    **until** $t_k > T$

---

Figure 4: Circle and three-leg structure with a size 4. State space is a 4-dimensional vector with each dimension as $\{0, 1\}$ representing whether the computer is working or not. Arrows represent the scopes of dimension. Each node has an arrow to itself, which we ignored in the figure.

## E Experiment Setups

**Circle and Three-leg structures.** Our computer network administrator domain with a circle and a three-leg structure Guestrin et al. (2001); Schuurmans and Patrascu (2002) are shown in Figure 4. Each computer gives a 1 reward when it is work and a 0 reward otherwise. The factored transition matrix for network with size $m$ is

$$P(s[i] = 0 \mid s[i] = 1, s) = \min\{1, \alpha_1 |\epsilon_i^1| + \sum_{j \in Z_i^P} \alpha_2 |\eta_{ij}^1| \mathbb{1}(s[j] = 0)\}, \ \forall i \in [m],$$

$$P(s[i] = 0 \mid s[i] = 0, s) = \min\{\max\{|\epsilon_i^0|, 0.5\} + \sum_{j \in Z_i^P} \alpha_2 |\eta_{ij}^0| \mathbb{1}(s[j] = 0)\}, \ \forall i \in [m],$$

where $\alpha_1, \alpha_2 = 0.1$ are constant and $\epsilon_i^1, \epsilon_i^0, \eta_{ij}^1, \eta_{ij}^0$ are all white noise. To avoid the extreme cases in our lower bound, both the MDPs are set to have limited diameters.