[Reviews · NeurIPS 2020]

Review 1

Summary and Contributions: This paper provides regret upper bounds for factored MDPs in the non-episodic setting. The bounds modify those of Osband and Van Roy 14, but are computationally efficient in that they require a weaker planning oracle. The bounds specialize to the UCRL2 bounds for non-factored, and additional bounds in terms of span are provided.

Strengths: The bounds appear no worse than UCRL2, and are computationally efficient. The discussion of the span as a better measure than diameter was helpful, and the non-efficient algorithm was also quite useful. Altogether, the results make an incremental but perhaps worthwile contribution to the study of factored MDPs.

Weaknesses: Since UCRL2, the dependences on states and actions have been improved somewhat. It would have been useful to understand if similar improvements can be attained here. Moreover, the techniques here seem quite standard, and it is hard to gauge the main technical accomplishment. Is it simply the observation that one does not need to restrict the MDP In the planning phase, thereby enabling better planning? This would have been useful to understand.

Correctness: While the proofs seem correct, they are *extremely* terse. The authors would do well to add additional exposition, and to explicitly cite results from other works they are using. This would be a tremendous help in gauging the correctness.

Clarity: The writing of the paper is generally clear, though I had difficulty locating the definition of h+

Relation to Prior Work: This past work does explain that its main advantage over osband and van roy is the weaker oracle. However, I would imagine there is much more recent work on factored MDPs that was not addressed. Moreover, the authors could have explained in greater detail (perhaps in an appendix) how the oracle assumption in the past work differs, and what the key insights were that enabled the weakening of this assumption. Update: My opinions remained unchanged, but my score would be substantially higher if the exposition were improved.

Reproducibility: Yes

Additional Feedback:


Review 2

Summary and Contributions: This paper proposes 2 new algorithms for solving factored MDPs with provable regret bounds. They also propose a new measure for the connectivity of factored MDPs and propose an approach for an optimal algorithm that matches a lower bound.

Strengths: The paper achieves better empirical performance than the best state of the art approach, factored RMAX. The introduction of factored span and characterization of an oracle method that meets the lower bound are important theoretical advances. Allows the factorization of both states and actions which is nice.

Weaknesses: The algorithms are only evaluated in 2 simple Admin domains. It is not clear how general are the results. FSRL algorithm needs a new oracle that corresponds to a bounded factored span. It is not clear what the computational tradeoff is going to be since it was not implemented. The approach assumes that the factored MDP structure is given.

Correctness: The claims appear to be correct, although I did not check the appendix carefully. The empirical methodology is fine.

Clarity: he readability suffers due to excessive notation and lack of expository text. The former may be hard to fix, but the authors can explain the ideas more clearly in text before introducing them. For example, the idea behind the construction of extended factored mdp should be clearer. The example in Figure 4 is useful, but why are we doing it? Why does it preserve the correctness? Why is this approach expected to do better than the previous constructions described in Section 3.1 and factored RMAX? Similarly Factored span may be illustrated by an example.

Relation to Prior Work: The paper can be clearer on its relationship to previous approaches and where its strengths are coming from. I have read the authors' response and am satisfied with it.

Reproducibility: Yes

Additional Feedback: pg 3. Rather than completely relying on notation, the authors should use pictures and text to give more intuition. In particular, I suggest a picture illustrating a concrete factored MDP to introduce the notation, eg. as a DBN. This can be used to solidify the understanding of the notation, eg., does Z_i represent parents of i in a DBN? pg5. Where is the notation x^s introduced? pg7. A picture would help understand factored span better. pg8. "arbitrary larger" -> "arbitrarily larger"


Review 3

Summary and Contributions: The paper considers an online factored MDP problem in the non-episodic setting. They provide ways for constructing tractable confidence regions for the latent parameters as well as the MDPs in the factored setting, and the two proposed algorithms scales only as the maximum size of a factor instead of the whole action or state space. The authors also discuss the notion of diameter and span in the factored MDP models, and provide a refined bound based on the sum of the factor spans.

Strengths: To my knowledge, factored MDPs in online and non-episodic settings was not studied, and I feel that the paper has quite a few novel contributions on handling the computational and learning issues in the factored models. For example, the construction of the factored set \tilde{X}, which contains only LW many elements in each of the factors, for the extended MDP M_k is new to my understanding. The discussions about the diameter and the span of factored MDPs, and Theorem 4 are also novel.

Weaknesses: - Theorem 2 does not seem to be precisely stated. For the Bayesian analysis to work in (Ouyang et al. 2017), we need to assume that the prior is supported on the set of (weakly)-communicating MDPs with diameter \leq D. From my own reading, I believe that this assumption is also needed for the Bayesian result, and the authors should clarify it. - While the submission is primarily theoretically driven, the paper could be made more attractive by providing an explicit application of the factored MDP model. - The authors should compare lower bound in Theorem 3 with the regret lower bound by (Jaksch et al. 2010). Restricting to the tabular case (i.e. when the factorizations are singleton), it seems that the resulting lower bound is weaker than (Jaksch et al. 2010).

Correctness: The paper is correct to my knowledge.

Clarity: The paper is clearly written for most parts, and here are some minor suggestions to polish the flow: - In Line 128, it is not quite clear what "adding extra discrete actions" means. - In Line 131-132, by saying that "by removing the sampling part", do you actually mean that you construct the discrete set of MDPs in a different way from sampling? - In Line 135, the authors could use a different letter from "W" to define the error bound, since "W" is already used in Line 105.

Relation to Prior Work: Relevant works are clearly discussed, and the related works are all included to my knowledge.

Reproducibility: Yes

Additional Feedback: To improve the paper, the authors should address the points raised in the "Weakness" box. #### Post Rebuttal ##### I have read the authors' feedback and other reviewers' opinion, and I maintain my overall score of accept.


Review 4

Summary and Contributions: The paper deals with FMDPs and provides regret bounds for DORL, PSRL and their own algorithm of FSRL.In addition, they provide a new measure for the connectivity of factored MDPs by extending the notion of span.

Strengths: New bounds for the non-episodic FMDP case are given. FMDPs are rarely considered despite having great potential in understanding and solving real world RL problems, and any work on that topic is welcomed.

Weaknesses: I find the work to be very problematic. The results I am familiar with regarding FMDPs usually substitute the size of the state space in MDP results with that of the effective state space given as the sum over possibilities for each factor. The results in this paper neglect this common view, but are given in context of either diameter (which can be huge in any problem where FMDPs are considered) or the notion of bias vectors of which I have little understanding of. The authors neglected to give any intuition regarding its meaning in regular MDPs, and specifically in their new described factored bias. Subsequently, I'm afraid the bounds that include the diameter are probably entirely irrelevant in FMDP setting squashing the inner understanding of the particular structure we're trying to exploit. The bounds that include the bias are on the hand unclear (at least for me) and give no intuition about the hardness of the general problem vs. that of the factored problem. While to the best of my knowledge the work is novel (might be due to the forgotten niches of FMDP and average reward setup), I am not convinced in its significance.

Correctness: I did not see any errors in the proofs. I'm a bit puzzled as to why the authors propose a new algorithm but provide only empirical results on the pre-existing algorithms. It might be my personal feel, but I would argue that in this kind of paper dealing with obviously far from reality bounds empirical simulations are a waste of space, and I would have much more appreciated more insight on the bounds and better explanations all around.

Clarity: I had a very hard time understanding the paper. 1. FMDP notation is usually very confusing going through the size of the alphabet, the number of factors, the size of the scope of each, and every sub-index. This paper was no help in getting these definitions straight. 1.a. Double notation: W is both an error and a bound on the size of the alphabet. bold h can receive an MDP (line 79), or a state (line 80). 1.b. The connection between S_i and X_i is confusing, taking the action out of X_i would have probably make things easier. 1.c. I have not seen the definition of h^+. 2. 3 algorithms were given by name, but none were given clear description. 3. Bulks of text that are defined but not used: lines 83-86, 107-112. 4. Typos - line 79 MPD -> MDP, line 196 l -> L. line 190 S_1 -> S_i. 5. The definition of the factored span is unclear, nor is the preposition 1 that follows it and whether its trivial or not.

Relation to Prior Work: Yes.

Reproducibility: Yes

Additional Feedback:


Review 5

Summary and Contributions: The paper investigates RL under the average-reward criterion in factored MDPs (FMDPs). The paper presents three algorithm for this setup: two oracle-efficient algorithms (PSRL and DORL), and FSRL which is not oracle-efficient. The paper provides a Bayesian regret bound for PSRL, as well as frequentist regret bounds for DORL and FSRL. It also presents a regret lower bound for FMDPs, established for product FMDPs.

Strengths: The paper is amongst very few papers studying RL in FMDPs with guarantees in terms of regret or sample complexity. To the best of my knowledge, this is the first work presenting computationally efficient algorithms for FMDPs in the average-reward setting. Also the presented regret lower bound for FMDPs is nice and new, and provides good insights into the intrinsic difficulty of the studied problem.

Weaknesses: I have some concerns and questions: - In order to come up with an efficiently-implementable algorithm, for DORL the authors construct an optimistic MDP following a very simple construction. This construction only considers error bounds and completely ignores the value function. So, while the proof claims the optimism is guaranteed, I believe that the resulting optimistic MDP is overly-optimsitic, and to favor computational efficiency, this way one may sacrifice learning efficiency to a large extent. Indeed, the idea of optimizing over a finite set of MDPs (in lieu of the bounded-parameter MDP) is nice. However, I believe the current construction that completely ignores the value function is too naive to work in practice. - The paper suggests to implement the algorithms with fixed-length (internal) episodes of length O(\sqrt{T/L}). This solution may only work for a fixed T, whereas in practice in many cases T is indefinite. Therefore, the stopping criterion used in the algorithms may need modifications. The stopping criterion used here could also be criticized from the standpoint of empirical performance: In early episodes, uncertainty is high and under any algorithm, it makes more sense to have shorter episodes as it is likely that any optimistic mode (MDP) would substantially differ from the true model. This observation is not however supported by the current algorithm design, and this could easily lead to a regret (empirically) suffering from a long linear phase. Finally I would like to add that the computational cost here grows as O(\sqrt T), whereas in UCRL2, the cost grows as O(C\log(T)). Overall, the reason of choosing such a stopping criterion is completely unclear from the main text. - In simulations, the reason behind choosing the hyper-parameter c is completely unclear, and this is indeed very strange. Did it happen to tune the regret? DORL is proven here to have a sublinear regret with the coefficients expressed in Eq. (2) and (3), and violating these constants could lead to anti-concentration, and thus, linear regret.

Correctness: The problem setup makes perfect sense to me. I was unable to check all the proofs given the limited review time. The regret bounds appear correct to me. However, I have some concerns regarding the validity of some parts of the experiments (see the previous section).

Clarity: The paper is overall well-written. There are some typos listed at the end of the review.

Relation to Prior Work: The authors have cited some relevant papers to FMDPs and learning in FMDPs. However, the literature review on regret minimization in tabular MDPs is missing, though these papers are indirectly related to this work.

Reproducibility: Yes

Additional Feedback: Some unclear statements: l. 167: We consider a simple FMDP …. but still solvable ===> the phrase “but still solvable” is not clear here. l. 112: “episode k”: Until this point, the notion of episode is not well-defined. It would be very helpful if you further state that the algorithms for average-reward RL typically proceed in internal episodes, where no reset takes place at the end of an episode. - About FSRL: When presenting FSRL to further tighten the regret, it is necessary to stress that it requires an additional knowledge of an upper bound on the span. Some typos: l. 29: the oracle planning oracle ===> the planning oracle-algorithms l. 76: … i.e., the diameter of … ===> would be more precise to write “for which the diameter of ….” l. 79: … MPD ===> MDP l. 92: action set ===> action space --- for consistency l. 98: there exists distributions ===> exist l. 101: there exist some ===> exists l. 112: with t_k be … ===> with t_k being … between l. 137 and 138: Let … be ..., be ===> grammatically incorrect l. 141: no verb in the first sentences l. 181: T step ===> steps Definition 5: of it optimal policy ===> … its optimal … l. 228: Our DORL and PSRL has ===> have l. 232: DORL and PSRL was ===> were l. 240: arbitrary larger ===> arbitrarily larger l. 242: Q equals to … ===> equals

[Author Response · NeurIPS 2020]

We thank all the reviewers for the constructive suggestions. To address concerns on readability, we included two extra
figures illustrating the factored MDPs and the factored span. We could not include them in the response due to the page
limitation. All the clarity questions are addressed in the main paper as well. This author response consists of two parts:
1) a table comparing computational oracles and regret bounds for different algorithms to clarify our main contributions;
2) point-to-point responses to questions from each reviewer.

**Comparing computational oracles and regret bounds.** To clarify our contributions, we provide the following table
comparing oracles and regret bounds for the algorithms mentioned in the paper. Our proposed DORL eases the oracle
in UCRL-Factored (adapted to non-episodic setting). FSRL is proposed for a tighter regret bound.

| Algorithms | F-RMAX | UCRL-Factored | PSRL | DORL | FSRL |
|---|---|---|---|---|---|
| Works | Strehl (2007) | Osband et al. (2014) | Osband et al. (2014) | This work | This work |
| Regret | (mixing rate)$T^{3/4}$ | $DT^{1/2}$ | $DT^{1/2}$ (Bayesian) | $DT^{1/2}$ | $Q(h(M))T^{1/2}$ |
| Oracle | Planning oracle | Optimizing average reward within a confidence set | Planning oracle | Planning oracle | Optimizing average reward with bounded factored span |

**Response to reviewer 1.** UCRL3 (Bourel et al., 2020) improves the dependence on actions and states by restricting
the set of successor states of the state-action pair. We believe this construction is quite independent from the FMDP
construction and the similar adaptation can easily be applied. Recently, Tian et al. (2020) closed the gap on horizon $H$
in episodic case. However, the story is more complicated in non-episodic case as the horizon length $H$ always serves as
a tight bound on the connectivity.

The techniques are mainly adapted from Agrawal and Jia (2017); Osband and Van Roy (2014), while we point out two
technical accomplishments: 1) we provide a new way to bound Bayesian regret of PSRL in non-episodic setting using
the fact that simple deterministic horizons can also achieve a near-optimal regret bound (Lemma 4). In UCRL2 and
Ouyang et al. (2017), length of episodes are (partly) determined by the doubling trick and are random variables. 2) The
proof of Theorem 4 in Appendix G provides a tighter deviation bound using factored span, which we haven't seen in
any previous literature.

On the clarity question, $h^+$ simply denotes the optimal bias vector of FMDP in the lower bound construction.

**Response to reviewer 2.** Our simulation settings are from Guestrin et al. (2002), while we agree that more experiments
on general environments shall be tested in the followings works.

Since the accurate algorithm does not exist even for standard planing oracle, we discuss approximate algorithm for
FSRL here. One can simply add an extra constraint on factored span for the approximate algorithm in Guestrin et al.
(2003) and the optimization problem can still be written as a linear programming and be solved efficiently.

On the other question about DORL, extended factored MDP is proposed to ease the oracle in UCRL-factored. The
correctness is preserved as the optimism can be preserved by only keeping a finite set of MDPs instead of a infinite set
of all the possible MDPs. Theoretically, DORL has an regret bound of $\sqrt{T}$, while factored RMAX only achieves $T^{3/4}$,
which is why we expect DORL to outperform factored RMAX.

**Response to reviewer 3.** Theorem 2 does not involve prior distribution as it considers the frequentist regret of DORL.
In Theorem 1, we explicitly pointed out that the true prior distribution $\phi$ is over the set of MDPs with diameters $\leq D$.

As mentioned above, we agree that more experiments on general environments shall be tests in the followings works.

Our lower bound seems weaker because we state the theorem in terms of span rather than diameter. However, we
suggest that when restricted to the tabular case, they are the same construction and have the same lower bound, because
in the construction of Jaksch et al. (2010), the span is propositional to the diameter.

In Line 128, "adding extra discrete actions" means for each sampled MDP, we add an copy of actions corresponds
to each of the FMDPs in the discrete set. In Line 131-132, instead of sampling, we directly construct a finite set of
FMDPs.

**Response to reviewer 4.** We used the simplest demonstration in the theorems. In fact, $m\sqrt{W}$ can be easily replaced by
$\sum_{i=1}^{m} \sqrt{|S_i|}$. We suggest that both diameter and span of bias vector are the most commonly used connectivity measure
in non-episodic MDPs and their intuitions have been well-understood in Puterman (2014); Bartlett and Tewari (2009).

On the questions about simulations, we proposed two new algorithms DORL and FSRL. While FSRL is not implemented,
DORL is tested in our simulation and achieved good regret curves. We also point out that the simulation indicates that
the optimal policy could still be found despite of using the approximate planner, which is of some significance.

[Meta-Review · NeurIPS 2020]

After discussing with the reviewers, we have decided to propose acceptance for the paper. Nonetheless, I would like the stress that the current submission has a number of critical aspects that need to be addressed by the authors to make the paper more solid. I strongly encourage the authors to read reviewers' comments and focus on improving along the following directions: - Clarity: Reviewers all agree that the paper could be improved in writing to make it more accessible to an audience that is not strictly familiar with the factored MDP formalism and/or the technicalities behind UCRL proofs. - The authors should further clarify the empirical results. In particular, it is unclear how the parameter c has been chosen and why it takes significantly different values for different algorithms. It would be helpful to see the performance as c changes. - The way optimism is obtained is probably not very tight and it may cause over exploration for a long time. This point should be discussed in much more detail. - The lower bound is an interesting novel result, but again it may require more discussing, in particular wrt to the non-factored case and why the span appears in this case and not in the non-factored case.